# Promotion Effects of *Taxus chinensis* var. *mairei* on *Camptotheca acuminata* Seedling Growth in Interplanting Mode

Chunjian Zhao [1,2,3,4], Sen Shi [1,2,3,4], Naveed Ahmad [5], Yinxiang Gao [6], Chunguo Xu [7], Jiajing Guan [1,2,3,4], Xiaodong Fu [1,2,3,4] and Chunying Li [1,2,3,4,*]

1 College of Chemistry, Chemical Engineering and Resource Utilization, Northeast Forestry University, Harbin 150040, China
2 Key Laboratory of Forest Plant Ecology, Ministry of Education, Northeast Forestry University, Harbin 150040, China
3 Engineering Research Center of Forest Bio-Preparation, Ministry of Education, Northeast Forestry University, Harbin 150040, China
4 Heilongjiang Provincial Key Laboratory of Ecological Utilization of Forestry-Based Active Substances, Northeast Forestry University, Harbin 150040, China
5 Department of Chemistry, Division of Science and Technology, University of Education, Lahore 54660, Pakistan
6 Institute of Jiangxi Oil-Tea Camellia, Jiujiang University, Jiujiang 332000, China
7 Dasuhe Forest Farm, Qingyuan, Liaoning 113312, China
* Correspondence: lcy@nefu.edu.cn; Tel./Fax: +86-451-8219-1387

**Abstract:** Wild *Camptotheca acuminata* Decne (*C. acuminata*) resources are becoming endangered and face poor growth. Preliminary investigation results found that the growth of *C. acuminata* in an artificial mixed forest of *Taxus chinensis* var. *mairei* (Lemee *et* Levl.), Cheng *et* L. K. Fu (*T. chinensis* var. *mairei*) and *C. acuminata* was significantly higher than that in pure forests. Understanding the reasons for the above differences can help create a mixed forest of *T. chinensis* var. *mairei* and *C. acuminata* to solve the problem of depleting *C. acuminata* resources. In this study, the growth and soil indexes under two different modes (*C. acuminata*/*T. chinensis* var. *mairei* interplanted and monocultured *C. acuminata* seedlings) were compared. The results showed that plant height, basal diameter, photosynthesis rate and chlorophyll content of *C. acuminata* under the interplanting mode were higher than those under monoculture. The growth rates of plant height and basal diameter that were calculated from interplanted specimens increased by 25% and 40%, respectively, compared with those from specimens that were monocultured. Photosynthetic rates from different light intensities under interplanting were higher than those in seedlings under monoculture. The contents of chlorophylls a and b and total chlorophyll under interplanting were 1.50, 1.59, and 1.47 times higher than those under monoculture, respectively. The numbers of bacteria and fungi in the interplanted culture were higher than those in the monoculture. Furthermore, the differences in microbial diversity under different planting modes were analyzed via the amplicon sequencing method. Soil enzyme activities increased under interplanting compared with that in the monoculture. Taxane allelochemicals were detected in the range of 0.01–0.67 μg/g in the interplanting mode from April to September. *T. chinensis* var. *mairei* may increase the establishment and productivity of *C. acuminata* seedlings under interplanting mode through improvements in enzyme activity, changes in microorganism population structure, and release of allelochemicals.

**Keywords:** *Camptotheca acuminata* Decne; *Taxus chinensis* var. *mairei* (Lemee *et* Levl.); Cheng *et* L. K. Fu; interplanting; allelochemicals; enzyme activity; microorganism

## 1. Introduction

Plantation management usually involves single cultivation, leading to reduced productivity and reforestation problems. It is highly desirable to develop specific techniques to

minimize and overcome the problems of managed plantations. One of the most developed techniques is using a mixed plantation rather than monocultures. Successful interplanting modes, based on carefully designed species mixtures, reveal many potential advantages in long-term practices [1]. The interspecific relationship among trees is one of the important factors that affect the survival of artificial mixed forests. Plant interactions are one of the important research topics in interspecific relationships, where plants can sense and recognize coexisting conspecifics or heterospecifics and thus adjust their growth, reproduction, and defense strategies [2]. Throughout its life cycle, a plant can interact simultaneously and sequentially, directly or indirectly, with many plant neighbors, whether in forests or in more natural environments. The nature and intensity of plant interactions are important factors that affect t plant populations and community structure, and the nature and intensity of plant interactions are influenced by abiotic/biotic environmental factors. This includes competition with heterospecifics (different species), reciprocal helping (i.e., mutually beneficial interactions), commensalism (i.e., facilitation) and asymmetric interactions such as parasitism and allelopathy [3]. The mechanisms of plant interactions have been investigated mainly from above-ground plants and below-ground soils [4]. Given the myriad of interactions between above-ground and below-ground communities and their well-known impacts on ecosystem function [5], it is often assumed that the composition of below-ground bacterial communities, allelochemicals, soil nutrients, enzyme activity, and above-ground plant communities will reflect one another [6–8]. Recently, studies have shown that some tree species have synergistic promotional effects. The intercropping with garlic promoted cucumber plant growth and attenuated damage caused by soil sickness [9]. As well as their promotion potential, Brassica species can be utilized to achieve higher productivity by using them as cover crops, companion crops, and intercrops for mulching and residue incorporation or simply by including them in crop rotations [10]. Therefore, it is useful to address poor forest productivity from the perspective of interspecific promotion.

*Camptotheca acuminata* Decne (*C. acuminata*) is a second-class key protected tree species in China. Camptothecin (CPT) is an important secondary metabolite in *C. acuminata*, and it has been used in the treatment of cancer [11]. The global demand for CPT antitumor drugs is increasing year by year, and its economic value is high. The process of isolating CPT from *C. acuminata* has become particularly important [12,13]. *C. acuminata* is used as a raw material for anti-cancer drugs; however, the unreasonable exploitation of wild-type *C. acuminata* has significantly reduced the distribution of wild resources [14,15]. Currently, researchers are focusing on the cultivation of *C. acuminata* plantations to alleviate the resource crisis of *C. acuminata* [16]. However, the poor growth of pure *C. acuminata* forests and the small amount of fruit per plant makes it necessary to choose the appropriate mixed tree species for cultivating *C. acuminata* mixed forests [15]. Researchers have found that *C. acuminata* growth can be improved by interplanting it with *Taxus chinensis* var. *mairei* (Lemee *et* Levl.) Cheng *et* L. K. Fu (*T. chinensis* var. *mairei*). The plant height and basal diameter of *C. acuminata* under interplanting are always higher than that of pure planting, and the average growth rate of the basal diameter in pure species of *C. acuminata* plantations is only 29.4% of that in mixed-species plantations [17]. Wang Jikun and other researchers improved the soil conditions of the mixed-planting mode among three species, *C. acuminata*, *T. chinensis* var. *mairei*, and Rosmarinus officinalis, and compared the average maximum photosynthetic rate of *C. acuminata* with that in pure planting mode; it was $16.95 \mu mol \cdot m^2 \ s^{-1}$, higher than that the planting mode of *C. acuminata* pure species [18]. Therefore, *T. chinensis* var. *mairei* has been advocated for interplanting in mixed *C. acuminata* mode, and it has become a successful interplanting strategy [17]. However, the potential mechanism involved in the interspecific relationship among the trees needs to be further studied.

*Taxus chinensis* var. *mairei*, which is listed as a first-class protected species, is an evergreen arbor that is ubiquitous in the southern region of China [19]. Previous studies have shown that the interplanting of *T. chinensis* var. *mairei* and *C. acuminata* can promote *C. acuminata* growth, which was explained from the perspective of improving the

microclimate of mixed forest land [20]. Our previous studies have found that cephalomannine, 10-deacetylbacatin III, paclitaxel, and 7-Epi-10-deacetyl-paclitaxel (7-Epi-10-DAT) in *T. chinensis* var. *mairei* were identified as allelochemicals that play a promotion role on *C. acuminata* seedlings [21]. This study measured the content of organic carbon, total nitrogen, the number of microbial populations, and enzymatic activities in soil under interplanting and monoculture in order to determine the effect of *T. chinensis* var. *mairei* on *C. acuminata* seedling growth from the perspectives of microbial population structure and soil chemical properties. The objective was to provide further theoretical support for interspecific relationships in the interplanting of *T. chinensis* var. *mairei* and *C. acuminata*.

## 2. Materials and Methods

### 2.1. Pot Experiment

This experiment was conducted in a greenhouse (45°43′9″ N, 126°38′5″ E) from April to September 2018 at the Key Laboratory of Forest Plant Ecology, Ministry of Education, Northeast Forestry University, China. The soil was collected from Xialian village, Xukou town, Hangzhou City, Zhejiang Province, China, and the soil type was yellow soil. *T. chinensis* var. *mairei* (5-year) and *C. acuminata* seedlings (1-year) were cross-interplanted at a 1:1 ratio in pots (40 × 40 × 23 cm, 50 kg of soil) in April. The plant spacing was kept to 20 ± 1 cm. *C. acuminata* seedlings under monoculture mode were used as control (Figure 1). Tap water was irrigated to each pot at the start, and the pots were kept moistened throughout the experimental period. All pots were randomly placed in the greenhouse, and plants were grown under day/night temperatures of 23/18 °C, with a relative humidity of 70%–75%, and a 12-h photoperiod. The growth index of the *C. acuminata* seedlings was measured within 1–3 days of the beginning of each month. Each treatment had 30 biologically effective replications.

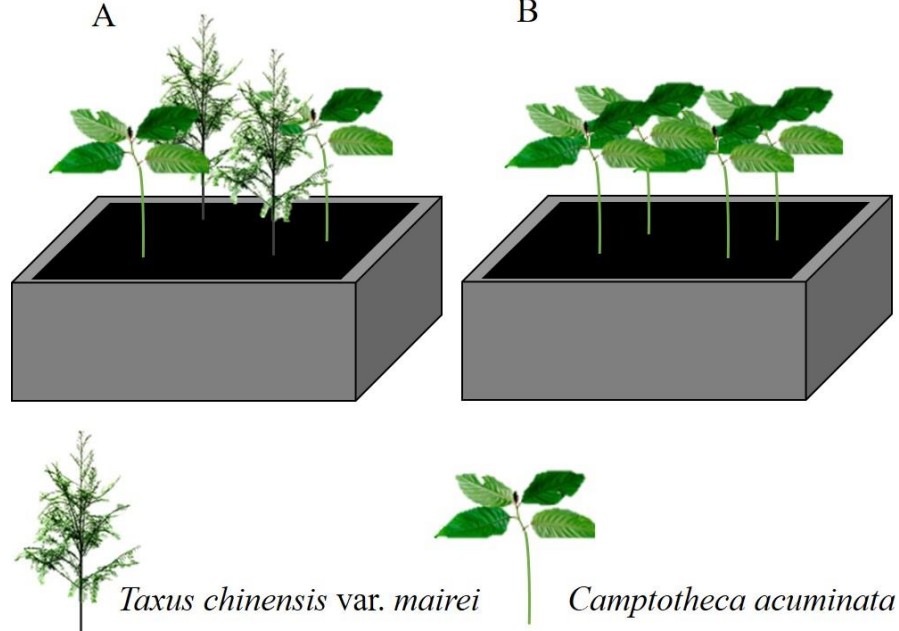

**Figure 1.** Diagram of the experimental planting modes. (**A**). Interplanted *C. acuminata* and *T. chinensis* var. *mairei*; (**B**). Monocultured *C. acuminata*.

### 2.2. C. acuminata Seedlings Bioassay

#### 2.2.1. Determination of Plant Height and Basal Diameter

The plant height was measured with a tape measure (units: cm; precision: 0.1 cm), and the basal diameter was measured with a vernier caliper (units: mm; precision: 0.02 mm) (Pulisen Measuring Tools Co. Ltd., Harbin, China).

2.2.2. Determination of Main Photosynthetic Indicators and Chlorophyll Content

The LI-6400 portable photosynthetic system (LI-COR Inc., Lincoln, NE, USA) was used to test the photosynthetic characteristics of the plants in different cultivation modes (interplanted *T. chinensis* var. *mairei*/*C. acuminata*; monoculture *C. acuminata*). The leaves were collected from the third leaf at the top of a *C. acuminata* plant. This plant material was cleaned, shade-dried, and stored in a freezer at $-20\,^\circ$C for further analysis. Chlorophylls of tissue samples were extracted with 80% acetone and determined using the Sosnowski method [22]. The time each day to measure the photosynthetic parameters of plant leaves was from 9 to 11 a.m.

2.2.3. Determination of Total Nitrogen and Organic Carbon in Soil

Five soil samples were randomly collected from a depth of 5–15 cm in the pots. Organic carbon content in rhizosphere soil was determined according to the $K_2Cr_2O_7$–$H_2SO_4$ wet oxidation method [23]. Total nitrogen concentration was analyzed via a colorimetric method using a continuous-flow autoanalyzer (AutoAnalyzer III, Bran, Luebbe GmbH, Germany) after the samples were digested with the Kjeldahl method [24].

2.2.4. Dynamic Changes in Soil Microbial Population and Enzyme Activities

Five soil cores (5–15 cm depth) were taken from each plot and thoroughly mixed. The samples were sieved (20-mesh) to remove plant residues and then homogenized. A sample of soil was stored at $-80\,^\circ$C immediately after soils were brought to the laboratory for determination of the soil parameters. The composition of the soil microbial population was evaluated using plate colony-counting methods [25,26]. All microorganisms were cultured at 28 $^\circ$C.

Soils were sampled in August 2018. Soil DNA was extracted from 0.4 g of frozen soil using the FastDNA™ SPIN Kit (MP Biomedicals TMInc, Eschwege, Germany), following modified according to manufacturers' recommendations [27]. Aliquots (50 μL) of DNA extracts were purified with the OneStep™ PCR Inhibitor Removal Kit (Zymo Research Biotech, Orange County, California, USA) and subsequently quantified using a microtiter plate assay with Quant-iT™ PicoGreen® dsDNA Reagent (Thermo Fisher Scientific, Bremen, Germany). DNA templates of each sample were prepared by diluting the purified DNA to 10 ng μL$^{-1}$ with nuclease-free water (Carl Roth, Roth-sur-Auhl, Rhineland, Germany). The 515 F 5′-barcode- (GTGCCAGCMGCCGCGG)-3′ and 5′-CTTGGTCATTTAGAGGAAGTAA-3′ primers were used to amplify 16 S rRNA genes at the hypervariable V4 regions for the bacterial and internal transcribed spacer 1 (ITS1) region. Then, they were amplified via polymerase chain reaction (PCR) via multiplexed barcoded amplicon sequencing by BMKCloud (www.biocloud.net) (accessed on Dec 10, 2022) on the Illumina MiSeq platform (Illumina, San Diego, CA, USA), as described by Castano et al. (2020) [28].

All five enzymes' (urease, sucrase, protease, phosphatase, and dehydrogenase) activities were determined via the colorimetric method [29]. The controls used distilled water instead of substrates. Each enzyme measurement had five replicates.

*2.3. Quantitation of Allelochemicals in Rhizosphere and Non-Rhizosphere Soils*

The quantitation of allelochemicals in interplanted *C. acuminata* and *T. chinensis* var. *mairei* rhizosphere soils and non-rhizosphere soils was completed via the UPLC-MS/MS method developed by our team [30].

*2.4. Statistical Analysis*

The experiments were carried out via a completely randomized design. Data were reported as the mean, standard error (S.E) of five replicates. For pot trials and the *C. acuminata* seedlings bioassay, SPSS software version 20.00 was used for statistical analysis. A student's *t*-test was used to evaluate differences among means. Differences were considered statistically significant at $p < 0.05$.

## 3. Results and Discussion

### 3.1. Effect of Interplanting on Plant Height and Basal Diameter of C. acuminata Seedlings

The heights and basal diameters of *C. acuminata* seedlings through interplanting (*T. chinensis* var. *mairei/C. acuminata*) in pot experiments on different days were measured. The results showed that *T. chinensis* var. *mairei* significantly increased *C. acuminata* seedling height and the basal diameter under interplanting mode; peaking appeared in September. From June to September, the heights and basal diameters of interplanted *C. acuminata* seedlings were all significantly higher than those under the *C. acuminata* monoculture (Tables 1 and 2). In September, the plant heights of the monoculture *C. acuminata* seedlings and interplanted seedlings increased by 95% and 120%, respectively, compared with April. The growth rate of plant height calculated from the interplanted mode increased by 25% compared with that from the monoculture mode. In September, the basal diameters measured from the monoculture and interplanted *C. acuminata* increased by 47% and 66%, respectively, compared with April. The basal diameter from the interplanted specimens increased by 19% compared with those from the monoculture. Our findings are consistent with the trends of previous studies. Kong's studies indicated that interplanting could effectively improve the growth of some plants. It has been observed that *J. mandshurica* establishment and management can be improved by interplanting with larch (*Larix gmelini*). The height, diameter at breast height, individual tree volume, and the stand stocking per hectare of *J. mandshurica* in the interplanting mode were 1.33, 1.87, 4.71, and 1.69 times higher than those in the monoculture mode, respectively, indicating that *L. gmelini* accelerates the growth of *J. mandshurica* in the interplanting mode [31]. *Pinus sylvestris* var. and *Broussonetia papyrifera* Linn. could be used to promote the growth of *A. pedunculata* seedlings, as well as enhance the construction of mixed plantations in coal mine degradation areas [32]. In our experiment, the interplanting of *C. acuminata* seedlings and *T. chinensis* var. *mairei* can significantly increase the growth rate of *C. acuminata* seedlings, and the positive effects of plant allelopathy depend to a large extent on the selection of mixed species.

**Table 1.** Effect of *C. acuminata* seedling height in different planting modes at different months.

| Planting Mode | Plant Height (cm) | | | | | |
|---|---|---|---|---|---|---|
| | April | May | June | July | August | September |
| MC | 50.1 ± 1.2 | 55.9 ± 2.3 | 73.9 ± 2.6 | 92.1 ± 2.4 | 95.7 ± 2.4 | 97.9 ± 2.5 |
| IC | 49.8 ± 1.1 | 56.7 ± 2.1 | 80.9 ± 2.3 * | 105.8 ± 2.6 * | 108.3 ± 2.6 * | 109.8 ± 2.7 * |

Each value is the mean value ± SE (*n* = 5); * Significant differences in different planting modes at these months (*t*-test, *p* < 0.05); MC, monoculture *C. acuminata*; IC, interplanted *C. acuminata*.

**Table 2.** Effect of *C. acuminata* seedling basal diameter in different planting modes at different months.

| Planting Mode | Basal Diameter (mm) | | | | | |
|---|---|---|---|---|---|---|
| | April | May | June | July | August | September |
| MC | 7.04 ± 0.04 | 7.23 ± 0.32 | 8.73 ± 0.44 | 9.04 ± 0.36 | 10.12 ± 0.32 | 10.33 ± 0.44 |
| IC | 7.02 ± 0.06 | 8.09 ± 0.34 * | 9.23 ± 0.42 * | 10.84 ± 0.42 * | 11.36 ± 0.38 * | 11.65 ± 0.56 * |

Each value is the mean value ± SE (*n* = 5); * Significant differences in different planting modes at these months (*t*-test, *p* < 0.05); MC, monoculture *C. acuminata*; IC, interplanted *C. acuminata*.

### 3.2. Effect of Interplanting on Photosynthesis of C. acuminata Seedlings

Under different light intensities, the net photosynthetic rate of *C. acuminata* seedlings interplanted with *T. chinensis* var. *mairei* was higher than that of *C. acuminata* seedlings in monoculture (Figure 2A); the net photosynthetic rate increased by 28%, 33%, 27%, 38% and 29% from May to September, respectively. In August, the net photosynthetic rate of *C. acuminata* seedlings interplanted with *T. chinensis* var. *mairei* was the highest. Compared with the monoculture, interplanting increased the stomatal conductance of *C. acuminata* seedlings by 24%, 25%, 27%, 27%, and 28% from May to September (Figure 2B), respectively, while the stomatal conductance of *C. acuminata* interplanted and monoculture seedlings under 100% light intensities (1600 µmol m$^2$·s$^{-1}$) was lower than that under other light

intensities conditions (800 µmol m$^2 \cdot$s$^{-1}$ and 480 µmol m$^2 \cdot$s$^{-1}$, respectively). Under different light intensities, interplanting with *C. acuminata* seedlings all enhanced intercellular $CO_2$ concentrations compared with seedlings in monoculture (Figure 2C); however, the growth rates decreased with an increase in processing time, which were 115%, 98%, 83%, 93% and 36% from May to September, respectively; meanwhile, the intercellular $CO_2$ concentrations of *C. acuminata* interplanted and monoculture seedlings in September were the lowest. The transpiration rate of *C. acuminata* interplanted with *T. chinensis* var. *mairei* was lower than that of *C. acuminata* seedlings in monoculture (Figure 2D), which was similar to the trend in intercellular $CO_2$ concentrations, which increased by 75%, 59%, 44%, 36% and 38% from May to September, respectively.

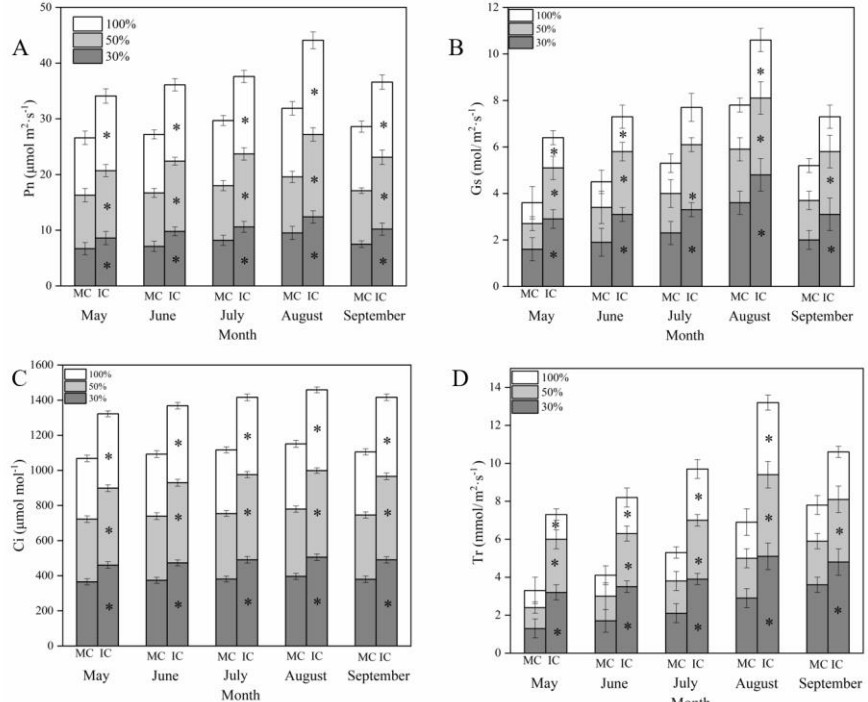

**Figure 2.** Effects of *C. acuminata* seedling-related indicators on the net photosynthetic rate (**A**), the stomatal conductance (**B**), the intercellular $CO_2$ concentrations (**C**) and the transpiration rate (**D**) in plant leaves under different planting modes at different months and light intensities. Each value is the mean value $\pm$ SE (*n* = 5). * Significant differences in different planting modes during these months (*t*-test, *p* < 0.05). MC, monoculture *C. acuminata*; IC, interplanted *C. acuminata*.

Light energy is the driving force of photosynthesis in green plants. In plant cultivation, rational utilization of light energy can help green plants fully photosynthesize [33]. Net photosynthetic rate, stomatal conductance, intercellular $CO_2$ concentration, and transpiration rate are important indicators of plant photosynthesis; they are closely related to plant growth and photosynthetic function. Interplanting (maize/soybean) decreased the net photosynthetic rate of soybean, but the rate recovered to varying degrees after the maize harvest [34]. In an interplanting system, two planting modes may affect the photosynthesis of each plant type. Our test showed that *T. chinensis* var. *mairei*/*C. acuminata* interplanting increased the net photosynthetic rate, stomatal conductance, intercellular $CO_2$ concentration, and transpiration rate of *C. acuminata* seedlings (Figure 2).

### 3.3. Effect of Interplanting on Chlorophyll Content of C. acuminata Seedlings

The chlorophyll content is an important indicator for evaluating plant growth. The contents of chlorophyll a, chlorophyll b, and total chlorophyll under different planting modes are shown in Figure 3. The chlorophyll content in the *C. acuminata* seedlings leaves under the different planting modes (interplanted *T. chinensis* var. *mairei*/*C. acumi-*

*nata*; monoculture *C. acuminata*) exhibited similar trends. The contents of chlorophyll a (Figure 3A), chlorophyll b (Figure 3B), and total chlorophyll (Figure 3C) were higher in the interplanted *C. acuminata* seedlings than those in the monoculture *C. acuminata* seedlings during the growth period. The content of chlorophyll was increased first and then decreased under different planting moods (interplanted *T. chinensis* var. *mairei*/*C. acuminata*; monoculture *C. acuminata*). Chlorophyll a, b, and total chlorophyll contents were 1.50, 1.59, and 1.47 times than those of the monoculture from May to September. The maximum content was obtained in July and August. The reason is possible that the soil in mixed planting contains more ingredients beneficial to plant growth, which promotes the growth of leaves and increases the chlorophyll content [35]. Chlorophyll in photosynthesis directly influences photosynthetic ability. The contents of chlorophyll in *C. acuminata* seedlings increased by interplanting with *T. chinensis* var. *mairei*, indicating enhanced photosynthetic rate leading to a higher biomass of *C. acuminata* seedlings. Thus, *T. chinensis* var. *mairei* exerts positive allelopathic effects on *C. acuminata* seedlings through allelochemicals in interplanting.

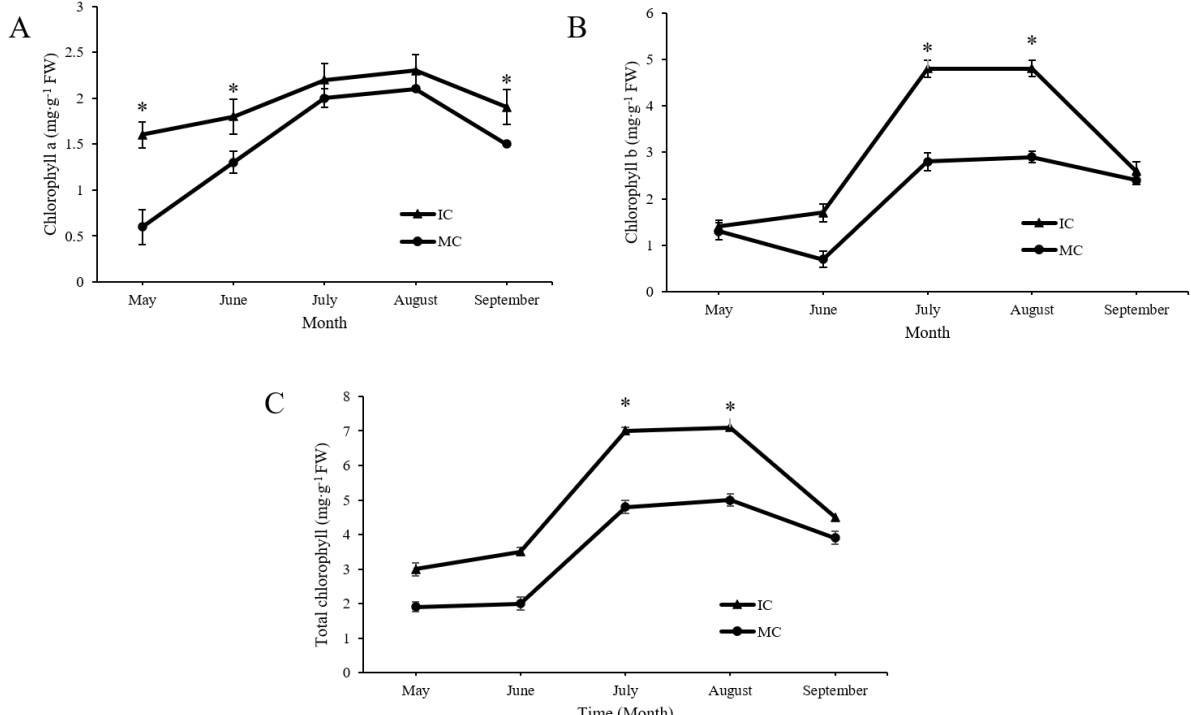

**Figure 3.** Effects on chlorophyll an (**A**) and b (**B**) content, and total chlorophyll content (**C**) *C. acuminata* seedlings under different planting modes at different months. Each value is the mean value ± SE (*n* = 5). * Significant differences in different planting modes during these months (*t*-test, $p < 0.05$). MC, monoculture *C. acuminata*; IC, interplanted *C. acuminata*.

### 3.4. Effect of Interplanting on Content of Organic Carbon and Total Nitrogen in Soil of C. acuminata Seedlings

The content of organic carbon and total nitrogen in both the monoculture and interplanted modes initially increased and then decreased during the growth period of the *C. acuminata* seedlings (Figure 4A,B). In the monoculture, the organic carbon level ranged from 17 to 24 mg·g$^{-1}$, with the lowest level occurring at the early stage of *C. acuminata* seedlings. In the interplanted mode, the organic carbon level was in the range from 19 mg·g$^{-1}$ to 32 mg·g$^{-1}$, with the lowest level occurring at the early stage of *C. acuminata* seedlings, and the organic carbon content in interplanted seedlings was higher than that in monoculture seedlings (Figure 4A). Similar to the results with organic carbon, variation in total nitrogen showed the same trends between the monoculture and interplanted treatments (Figure 4B). For the monoculture, the content of total nitrogen initially increased

from May to September, peaking at 8.0 mg·g$^{-1}$ in August. For the interplanting mode, the total nitrogen content increased gradually from May to August, peaked at 9.3 mg·g$^{-1}$, and then fell slowly after August.

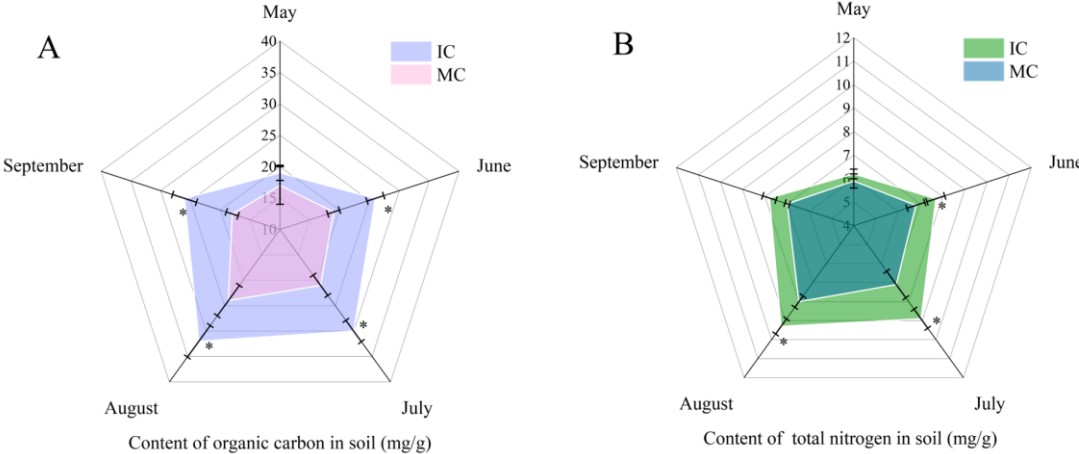

**Figure 4.** Effects on soil organic carbon (**A**) and total nitrogen contents (**B**) under different planting modes in different months. Each value is the mean value ± SE (*n* = 5). * Significant differences in different planting modes during these months (*t*-test, *p* < 0.05); MC, monoculture *C. acuminata*; IC, interplanted *C. acuminata*.

Sufficient but not excessive nutrient element levels in the soil are vital for *C. acuminata* seedling growth. The results showed that the contents of total nitrogen and organic carbon in the interplanted soil were higher than those in the monoculture soil. These nutrient levels were all-sufficient for *C. acuminata* seedling growth, and interplanting *T. chinensis* var. *mairei* with *C. acuminata* was helpful to avoid salinization of the soil under mixed cultivation and improve fertilizer use efficiency. Higher total nitrogen and organic carbon levels were observed in interplanted *C. acuminata* seedlings with *T. chinensis* var. *mairei*. These results are in almost agreement with those of Wei et al. [36] and indicate that interplanted *T. chinensis* var. *mairei* and *C. acuminata* seedlings play an important role in activating and balancing nutrients in the soil. We think that this is due to the fact that plants and soils have close interactions, such as the exchange of substances and nutrient cycling.

### 3.5. Effect of Interplanting on Enzyme Activity in the Soil of C. acuminata Seedlings

*C. acuminata* seedlings urease, sucrase, dehydrogenase, protease, and phosphatase activities between interplanted and monoculture modes from May to September were determined, and the results are shown in Figure 5. It can be seen from Figure 5 that the enzyme activity of the interplanted soil was significantly higher than that of the monoculture soil. In August, enzyme activity from the interplanted mode was significantly higher than that monoculture mode (*p* < 0.05).

Soil enzyme activity is an important indicator of soil quality. Generally, extracellular enzymes are secreted by soil microorganisms in order to decompose large, polymeric compounds and are closely related to the cycling of carbon, nitrogen, and phosphorus [37,38]. These results are in agreement with those of Zeng et al.'s research. These results showed that intercropping turmeric and ginger with patchouli could improve soil enzymes. They also modify the soil's physical and chemical properties through changes in enzyme activity [39]. Zhao et al. reported that higher activities of catalase, urease, invertase, and phosphatase were detected in maize/hot pepper interplanted soil [40]. Significantly higher activities of invertase, urease, and phosphatase in the soil of a garlic/cotton interplanting system throughout the course of growth in comparison to a cotton monoculture were observed. Interplanting garlic enhanced its phosphatase activity and cellulase activity [41].

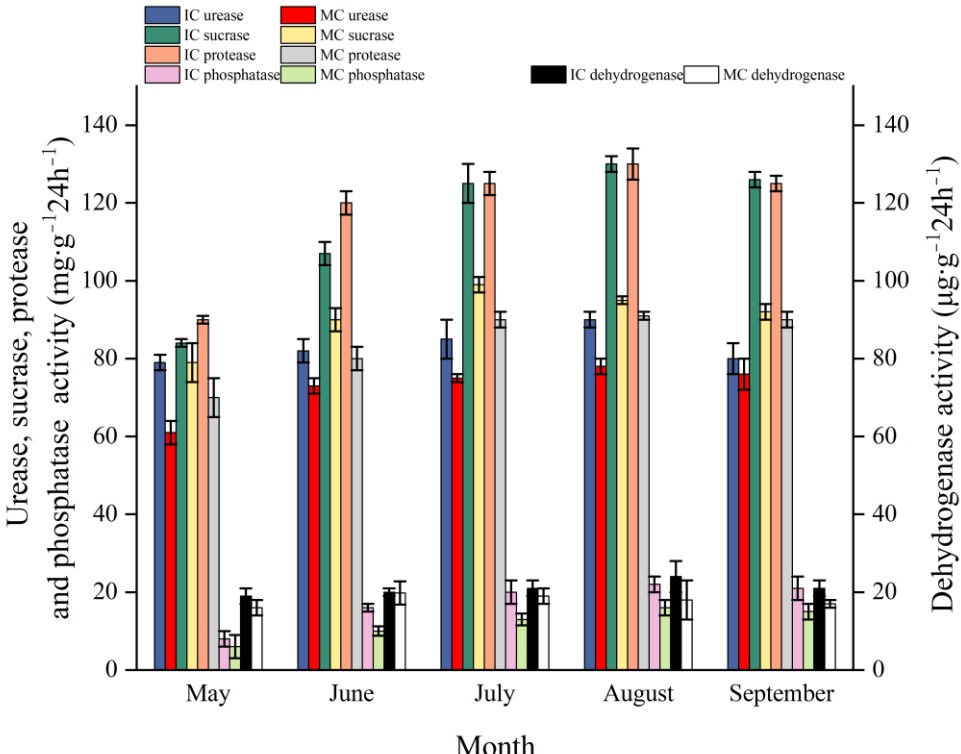

**Figure 5.** Effects on urease, sucrase, protease, and phosphatase (mg·g$^{-1}$ 24 h$^{-1}$), dehydrogenase (µg·g$^{-1}$ 24 h$^{-1}$) activities in soil under different planting modes at different months. Each value is the mean value ± SE (*n* = 5). MC, monoculture *C. acuminata*; IC, interplanted *C. acuminata*.

### 3.6. Effect of Interplanting on Microbial Populations in the Soil of C. acuminata Seedlings

Bacteria populations in soil appeared to change during the different months between the monoculture and interplanted *C. acuminata* seedlings. However, the bacterial population reached a sharp peak for both modes in August, and the bacterial population in the interplanted soil was significantly higher than that in the monoculture soil (27.6 × 10$^6$ > 19.6 × 10$^6$ cfu·g$^{-1}$) (Figure 6A). Fungal populations in the soil were significantly different between the monoculture and interplanted modes from July to September (*p* < 0.05), and the trend was similar between the monoculture and interplanted modes (Figure 6B). In the interplanted culture, the fungal population increased significantly to 11.9 × 10$^5$ cfu·g$^{-1}$ in September. In the monoculture, the number of fungi showed a similar trend to that in the interplanted mode each month and peaked in August at 7.51 × 10$^5$ cfu·g$^{-1}$.

The compositions of bacterial and fungal communities were tested for their taxonomy in interplanted and monoculture *C. acuminata* modes. The relative abundances of bacterial phyla Acidobacteriota (0.320), Proteobacteria (0.271), Methylomirabilota (0.104), Myxococcota (0.063), Actinobacteriota (0.044) and Gemmatimonadota (0.028) in monoculture *C. acuminata*, as well as the archaeal phylum Acidobacteriota (0.260) and Proteobacteria (0.270) in interplanted *C. acuminata*, encompassed the largest proportion of sequences of bacterial communities (Figure 6C). In interplanted *C. acuminata*, members of phyla Ascomycota (0.748), Basidiomycota (0.128), Chytridiomycota (0.029), and Glomeromycota (0.016) were prevalent fungal groups across the investigated soils. Unlike the interplanted *C. acuminata*, the monoculture *C. acuminata* showed a fungal composition of mainly Ascomycota (0.664), Basidiomycota (0.150), Chytridiomycota (0.051), Mortierellomycota (0.045), Glomeromycota (0.025) and Calcarisporiellomycota (0.012) (Figure 6D).

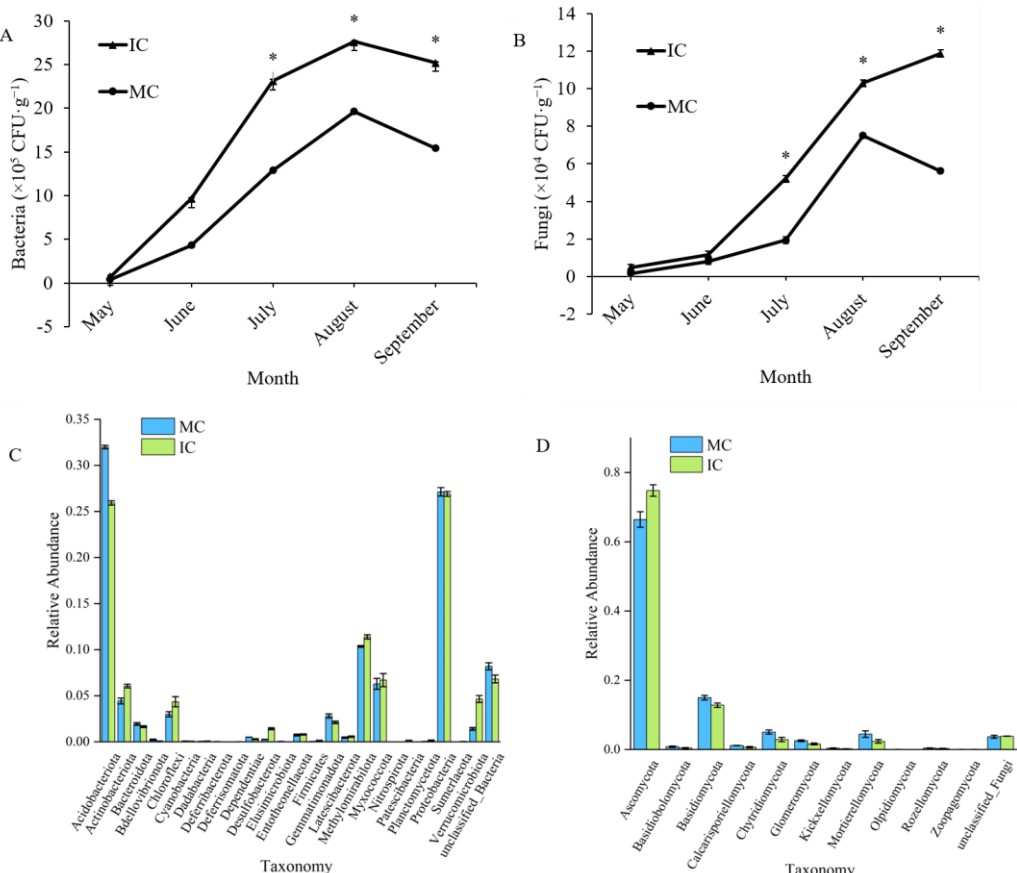

**Figure 6.** Effects on bacteria (**A**) and fungi (**B**) in soil under different planting modes at different months. * Significant ($p < 0.05$) differences in planting modes during these months (*t*-test). MC, monoculture *C. acuminata*; IC, interplanted *C. acuminata*. Relative abundance of bacterial (**C**) and fungal (**D**) communities at the phylum level, based on Illumina sequencing of the 16S rRNA gene and the ITS1 region from two plant modes (MC, monoculture *C. acuminata*; IC, interplanted *C. acuminata*). Each value is the mean value $\pm$ SE ($n$ = 3).

Microorganisms are one of the important components of the soil ecosystem. Soil microbial populations play an important role in many soil functions, including organism decomposition and nutrient cycling [42,43]. A previous study showed that interplanting might affect the number of soil microbial populations under continuously interplanted modes. Intercropping between sugarcane and soybean is widely used to increase crop yield and promote the sustainable development of the sugarcane industry. Intercropping improved the bacterial population in the rhizosphere in comparison with a monoculture [44]. Elshahawy et al. showed that interplanting cucumber with onion or garlic increased the population of bacteria [45]. Tian and Boparai et al. stated that wheat or garlic interplanting with cotton significantly enhanced the population of bacteria and actinomycetes in the rhizosphere compared to monoculture planting; garlic interplanting had even more remarkable effects [46,47]. Ali et al. reported that the soil population of bacteria and actinomycetes increased under a cucumber/garlic interplanting system compared to a cucumber monoculture [48]. Interplanting increased the number of rhizosphere fungi in aromatic plants [49]. In our study, similar results were also obtained, where interplanting *T. chinensis* var. *mairei* with *C. acuminata* seedlings in soil enhanced the populations of bacteria and fungi in comparison with monoculture *C. acuminata* seedlings. A comparison of the diversity of microbial composition in monoculture *C. acuminata* and interplanted *C. acuminata* showed that there were differences between them. It is very clear that plant–microbe interactions in the rhizosphere are influenced by several factors. As an indispensable energy and nutrient source for microorganisms, soil organic matter content was reported to play

an important role in shaping microbial communities [50]. Secondary metabolites in plant root secretions also affect the growth of particular microbes [51]. As several studies have established, the rhizosphere microbiome composition greatly affects plant growth, and plants employ several mechanisms to recruit specific microflora [52]. Our results fully demonstrated that the balance of microorganisms in the soil would not be destroyed under interplanting modes; moreover, the reasonable microbial population distribution played a role in promoting the growth of *C. acuminata.* It has been shown that methylomirabilota can enhance soil carbon and nitrogen cycling to a certain extent, and ascomycota is significant in promoting plant growth [53]. In this study, both methylomirabilota and ascomycota were found in higher percentages in the soil of interplanted *C. acuminata* than in the soil of monoculture *C. acuminata*; therefore, we infer that the better growth of interplanted *C. acuminata* comes from these two microbials, which may be promoting nutrient utilization and uptake, further affecting the growth of *C. acuminata*.

### 3.7. Release of Taxane Allelochemicals from T. chinensis var. Mairei in Interplanting Mode

All allelochemicals in the rhizosphere and non-rhizosphere soil were determined, and the results are shown in Table 3. The content of 10-DAB III peaked at 0.67 µg/g in rhizosphere soil and 0.21 µg/g in non-rhizosphere soil in August. In the same month, the content of 7-Epi-10-DAT was the lowest; only 0.32 and 0.12 µg/g were found in the rhizosphere and non-rhizosphere soil, respectively. The contents of cephalomannine and paclitaxel were less than 0.01 µg/g in both rhizosphere and non-rhizosphere soils. It is possible that soil microorganisms degrade the allelochemicals, or it may be related to the adsorption of the soil to them [54]. In addition, the physicochemical properties, release concentrations, environmental factors, and retention and transformation in the soil have an impact on whether allelochemicals are detected in the soil or not [55]. Therefore, there were two allelochemicals (10-DAB III and 7-Epi-10-DAT) in the rhizosphere and non-rhizosphere soil, which suggested that they may be present in the interplanting mode, which consequently affects the growth of *C. acuminata* seedlings.

The soil ecosystem is a complex composition in which soil enzymes and soil microorganisms play an important role in promoting the soil nutrient cycle [56]. The physical and chemical properties of soil have a great impact on soil enzyme activity and soil microbial community structure [57]. In contrast to competition for resources from soil, allelopathy involves the release of allelochemicals from plants into the environment [58]. Accordingly, the detection of allelochemicals from rhizosphere soil and non-rhizosphere soil is a key to understand plant–plant interspecific interactions [59]. A series of interactions between allelochemicals and soil abiotic and biotic factors may occur when allelochemicals are released through the soil [55]. In particular, soil microbial interactions radically alter the environment and provide a much better indication of real effects [60]. The presence of taxane medium allelochemicals may be one of the important factors for *T. chinensis* var. *mairei* to promote *C. acuminata* seedling growth under the interplanting mode. However, the interactions among allelochemicals, soil enzymes and soil microorganisms require further clarification.

**Table 3.** Contents of cephalomannine, 10-DAB III, paclitaxel and 7-Epi-10-DAT in rhizosphere soils and non-rhizosphere soil.

| Compounds (μg/g) | April | | May | | June | | July | | August | | September | |
|---|---|---|---|---|---|---|---|---|---|---|---|---|
| | Rhizosphere Soil | Non-Rhizosphere Soil | Rhizosphere Soil | Non-Rhizosphere Soil | Rhizosphere Soil | Non-Rhizosphere Soil | Rhizosphere Soil | Non-Rhizosphere Soil | Rhizosphere Soil | Non-Rhizosphere Soil | Rhizosphere Soil | Non-Rhizosphere Soil |
| Cephalomannine | $<0.01 \pm 0.01$ | $<0.01 \pm 0.01$ | $<0.01 \pm 0.01$ | $<0.01 \pm 0.01$ | $<0.01 \pm 0.01$ | $<0.01 \pm 0.01$ | $<0.01 \pm 0.01$ | $<0.01 \pm 0.01$ | $<0.01 \pm 0.01$ | $<0.01 \pm 0.01$ | $<0.01 \pm 0.01$ | $<0.01 \pm 0.01$ |
| 10-DAB III | $0.23 \pm 0.01$ | $0.09 \pm 0.01$ | $0.34 \pm 0.01$ | $0.12 \pm 0.01$ | $0.46 \pm 0.01$ | $0.15 \pm 0.01$ | $0.65 \pm 0.01$ | $0.20 \pm 0.01$ | $0.67 \pm 0.01$ | $0.21 \pm 0.01$ | $0.63 \pm 0.01$ | $0.19 \pm 0.01$ |
| Paclitaxel | $<0.01 \pm 0.01$ | $<0.01 \pm 0.01$ | $<0.01 \pm 0.01$ | $<0.01 \pm 0.01$ | $<0.01 \pm 0.01$ | $<0.01 \pm 0.01$ | $<0.01 \pm 0.01$ | $<0.01 \pm 0.01$ | $<0.01 \pm 0.01$ | $<0.01 \pm 0.01$ | $<0.01 \pm 0.01$ | $<0.01 \pm 0.01$ |
| 7-Epi-10-DAT | $0.13 \pm 0.01$ | $0.04 \pm 0.01$ | $0.17 \pm 0.01$ | $0.06 \pm 0.01$ | $0.22 \pm 0.01$ | $0.09 \pm 0.01$ | $0.30 \pm 0.01$ | $0.11 \pm 0.01$ | $0.32 \pm 0.01$ | $0.12 \pm 0.01$ | $0.31 \pm 0.01$ | $0.11 \pm 0.01$ |

## 4. Conclusions

The *T. chinensis* var. *mairei* and *C. acuminata* seedling interplanted cultivation resulted in significant positive effects and improvements in the establishment and productivity of *C. acuminata* seedlings and involved multiple factors. In this study, biomass, net photosynthetic rate, chlorophyll, organic carbon, total nitrogen content, microbial populations, and enzyme activities in the soil of interplanted *C. acuminata* seedlings increased. The change in soil microbial diversity under the interplanting of *C. acuminata* was found to be beneficial to the growth of *C. acuminata*. Such effects should be correlated to the joint action of microbial and enzymatic activities in the soil after interplanting with *T. chinensis* var. *maire*. Meanwhile, taxane allelochemicals released from the root exudate of *T. chinensis* var. *maire* into the soil in the interplanted culture may be one of the reasons for enhanced growth in *C. acuminata* seedlings. Thus, our study provides theoretical support for clarification of the promotional effects of *T. chinensis* var. *mairei* on *C. acuminata* seedling growth in interplanted mode. It also offers many potential implications and applications in managed tree ecosystems. However, how microorganisms enhance enzymatic activity or transfer nutrients and interact with allelochemicals to promote the growth of *C. acuminata* seedlings in interplanted soils are mechanisms that require further exploration.

**Author Contributions:** Conceptualization, C.Z. and S.S.; methodology, S.S.; software, S.S.; validation, C.L.; formal analysis, Y.G.; investigation, J.G.; resources, C.X.; data curation, S.S. and J.G.; writing—original draft preparation, S.S.; writing—review and editing, C.Z., C.L. and Naveed Ahmad; visualization, S.S.; super-vision, X.F.; project administration, C.L.; funding acquisition, C.Z. and C.L. All authors have read and agreed to the published version of the manuscript.

**Funding:** This work was financially supported by the Fundamental Research Funds for the Central Universities (2572022AW25), the National Natural Science Foundation (31870609), Science and Technology Program of Jiangxi Administration of Traditional Chinese Medicine (2020A0379), and the 111 Project, China (B20088).

**Data Availability Statement:** Data are available on request from the corresponding author.

**Conflicts of Interest:** The authors declare no conflict of interest.

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
