# Peer review of "Promotion Effects of Taxus chinensis var. mairei on Camptotheca acuminata Seedling Growth in Interplanting Mode"

_forests, doi:10.3390/f13122119_

Round 1
Reviewer 1 Report (Previous Reviewer 2)
I agree for the paper to be published in present form.
Author Response
Thank you.
Reviewer 2 Report (Previous Reviewer 1)
1. The introduction needs to supplement the research progress on the mechanism of plant interaction; 2. The statistical method description in the results section is inconsistent with the method section, please modify it; 3. Continue to check the language of the full text, not limited to the suggestions of the reviewers.
Author Response
Dear Reviewer,
We are so appreciated for your e-mail on December 1 2022, regarding the review results of our manuscript entitled “Promotion Effects of Taxus chinensis var. mairei on Camptotheca acuminata Seedling Growth in Interplanting Mode” (forests-2094530).
We are so thankful for the constructive comments, which would help us both in English and in depth to improve the quality of the manuscript. Here our manuscript has been revised carefully and rigorously according to the constructive comments from the reviewers. We hope that our manuscript could be considered for publication in your journal.
Sincerely yours,
Chunjian Zhao, Ph.D.
zcj@nefu.edu.cn
Response to the comments of the reviewer point-by-point is as follows:
Reviewer:
- The introduction needs to supplement the research progress on the mechanism of plant interaction.
Response: The mechanism of plant interactions has been supplemented in the introduction (see lines 46-61)
- The statistical method description in the results section is inconsistent with the method section, please modify it.
Response: Statistical methods have been modified (see 2.4. Statistical analysis in revised manuscript)
- Continue to check the language of the full text, not limited to the suggestions of the reviewers.
Response: The manuscript has been polished by the MDPI Official Language Service.
This manuscript is a resubmission of an earlier submission. The following is a list of the peer review reports and author responses from that submission.
Round 1
Reviewer 1 Report
1. Why did the interaction experiment of two tree species study the growth of only one species?
2. It is discussed in this study that the interaction mode promotes tree growth by affecting soil enzyme activity and microbial community structure. We know that soil nutrients have a more direct impact on plant growth, so why is this part not included in this study?
3. The writing of this paper is poor, such as poor picture quality, grammar, font errors, etc., which requires extensive modification. Please find the attachment for details.

Reviewer 2 Report
The title is suggestive and suitable for the content of this paper.
In the abstract, the purpose of the work is identified, but I think that the justification for the research should be specified more clearly and should be included in the abstract.
Introduction, methods and results are well organized, fact that shows the correct and detailed documentation of the authors, but I would replace figures 2 and 3 because they are not very clear and the content cannot be seen very well.
The conclusions are correctly supported by the results presented in the paper. Overall it looks like a good work.
I also recommend a short linguistic check.